# Quercetin Is a Novel Inhibitor of the Choline Kinase of *Streptococcus pneumoniae*

**DOI:** 10.3390/antibiotics11091272

**Published:** 2022-09-19

**Authors:** Tahl Zimmerman, Salam A. Ibrahim

**Affiliations:** Food Microbiology and Biotechnology Laboratory, Food and Nutritional Sciences Program, Department of Family and Consumer Sciences, College of Agriculture and Environmental Sciences, North Carolina Agricultural & Technical State University, Greensboro, NC 27411, USA

**Keywords:** antibiotic, antimicrobial, flavonoid, food-borne pathogens

## Abstract

The effectiveness of current antimicrobial methods for addressing for food-borne Gram-positive pathogens has dropped with the emergence of resistant strains. Consequently, new methods for addressing Gram-positive strains have to be developed continuously. This includes establishing novel targets for antimicrobial discovery efforts. Eukaryotic choline kinases have been highly developed as drug targets for the treatment of cancer, rheumatoid arthritis, malaria and many other conditions and diseases. Recently, choline kinase (ChoK) has been proposed as a drug target for Gram-positive species generally. The aim of this work was to discover novel, natural sources of inhibitors for bacterial ChoK from tea extracts. We report the first natural bacterial ChoK inhibitor with antimicrobial activity against *Streptococcus pneumoniae*: quercetin.

## 1. Introduction

The effectiveness of current antimicrobial methods for addressing for food-borne Gram-positive pathogens has dropped with the emergence of resistant strains. Consequently, new methods for addressing Gram-positive strains have to be developed continuously. This includes establishing novel targets for antimicrobial discovery efforts. Eukaryotic choline kinases have been highly developed as drug targets for the treatment of cancer, rheumatoid arthritis, malaria and many other conditions and diseases. Recently, choline kinase (ChoK) has been proposed as a drug target for Gram-positive species generally [1] after the first ChoK inhibitor capable of both blocking the activity of the ChoK *Streptoccus pneumoniae* (sChoK) and blocking the growth of *S. pneumoniae* cells was discovered [2]. Later, even more powerful sChoK inhibitors were elucidated [3]. ChoK inhibitors from natural sources have been discovered in the past [4], but so far, none have been discovered known to inhibit sChoK or any other bacterial isoform [5].

All ChoK isoforms phosphorylate choline (Cho) into phosphocholine (PCho). In the case of *S. pneumoniae*, PCho is incorporated into two types of teichoic acids: lipoteichoic acid (LTA) and cell wall teichoic acid (CTA) [6]. LTA and CTA are molecularly analogous. However, LTA is embedded in the cell membrane via a lipid anchor, and CTA is embedded in the cell wall peptidoglycan layer [7]. LTA is a known virulence factor [8], and the LTA synthesis pathway is a validated drug target [9].

The sChoK enzyme is an element of the pathway that mediates the decoration of teichoic acids with PCho via the intermediary CDP–choline. Genes that are part of this pathway are expressed via the *lic* gene locus. The *LicB* gene expresses a Cho transporter which collects Cho from the external environment. *LicA* codes for sChoK, which phosphorylates Cho into PCho. *LicC* is a gene coding for cytidylyl transferase, which converts PCho into CDP–choline. The LicD1 and LicD2 PCho transferases remove PCho from CDP–choline and attach them to *N*-Acetylgalactosamine (GalNAc) residues located on pre-teichoic acid glycan subunits [10]. These residues are then polymerized by an unidentified protein to form teichoic acid which is transported across the cell membrane acid flippase TacF [10]. The TacL ligase attaches teichoic polymers to a glycolipid anchor to form LTA [8]. Meanwhile, teichoic acid polymers are cross-linked to peptidoglycan to form WTA by the action of LCP phosphotransferases [10].

Using *S. pneumoniae* and a colorimetric method for detecting sChoK as models, we screened eight different soluble tea extracts for anti-ChoK and antimicrobial activity with the ultimate aim of discovering novel ChoK inhibitors capable of preventing the growth of Gram-positive bacteria. These teas are Espinheira Santa, Peppermint, Lemon Broth, Wormwood, Rose Hips, Rooibos, Hibiscus, and Saint-Johns’ Wort. These teas have been previously been identified as having antimicrobial activity, however their drug targets have not been described (See Table 1 below).

## 2. Results and Discussion

### 2.1. Hibiscus and Saint-John’s Wort Inhibit sChoK Activity and S. pneumoniae Cell Growth

Using a colorimetric method of detecting choline consumption, each tea extract (Table 1) was individually assayed for inhibitory potential against recombinant sChoK. While all teas displayed some level of inhibition, the three teas with the strongest inhibitory activity (as determined by % Inhibition) were RB, HB, and SJ. This result indicated that all teas contained some compounds capable of inhibiting sChoK, but that the teas with the highest concentration of these compounds were RB, HB, and SJ with SJ > HB > RB. Minimum inhibitory concentrations (MIC) were also measured for each tea in broth cultures. Interestingly, the MICs generally correlated with the % Inhibition, in that the teas with the strongest sChoK inhibition also had the lowest measured MICs (Table 1). This result suggested that the antimicrobial activity of these teas was due to the choline kinase inhibitors contained within the teas. The inhibition zones in the plating assay confirmed that the teas with the strongest inhibitory activity were RB, HB, and SJ, though here there was a discrepancy with the broth culture results: HB > SJ > RB (Figure 1A).

The antimicrobial activity of these teas has been reported in the past (see Table 1). However, this is the first time that an enzymatic target of these teas has been identified and suggested to be responsible for their antimicrobial activity.

### 2.2. Quercetin Is an Inhibitor of sChoK and Has Antimicrobial Activity against S. pneumoniae

Teas are known to contain many flavonoid compounds which function as active ingredients with antimicrobial activity. Our initial hypothesis was that at least one of these flavonoids was responsible for sChoK inhibition. One abundant flavonoid in the HB, SJ, and RB extracts is quercetin. Indeed, quercetin is the most abundant flavonoid in HB [18] and is known to have antimicrobial activity. Quercetin is also a known inhibitor of acetylcholine esterase, which suggested that this compound could be active against choline metabolism enzymes generally [19]. Colorimetric assays revealed that quercetin inhibited sChoK in a dose-dependent manner (Figure 1B) with an IC_50_ of 6.17517+/−3.62 µM. Meanwhile related flavonoids rutin, kaempferol, and kaempferol-3-rutinoside were far less effective against sChoK activity, indicating the specificity of quercetin antimicrobial activity. The antimicrobial activity of quercetin against *S. pneumoniae* was confirmed in broth assays, in which an MIC of 150 µM was measured for this flavonoid (see Figure 2).

**Figure 2 antibiotics-11-01272-f002:**
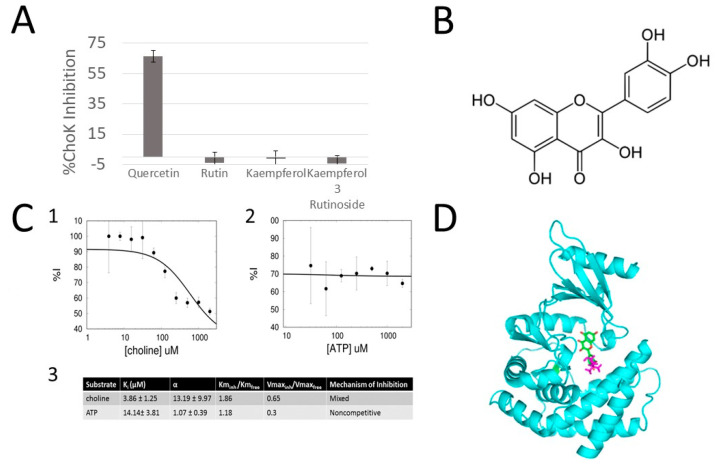
(**A**) Choline kinase enzymatic assays, in which the % inhibition of choline kinase is measured in the presence of 100 µM of several flavonoids. As seen in the figure, one flavanoid has a highly inhibitory effect against choline kinase while a few related flavonoids do not. (**B**) The chemical structure of the most potent flavonoid identified (**C**) kinetic study of quercetin versus ATP and choline substrates: (**1**) inhibition analysis of quercetin in competition with choline. (**2**) Inhibition analysis of quercetin in competition with ATP. (**3**) Kinetic constants demonstrating the mechanism of inhibition of quercetin versus choline and ATP, an α > 1 indicates competitive behavior, an alpha = ~1 is indicative of non-competitive behavior. Likewise, a Km(inh)/Km(free) > 1 is indicative of the competitive behavior of quercetin with respect to choline (**D**) Docking model of binding of quercetin (green) to the choline binding site of choline kinase (cyan). The choline molecule is shown in purple.

These results establish quercetin for the first time as a bona fide inhibitor of sChoK, the first ever discovered from a natural plant source in a screening process. In addition, this result further validates the strategy of using screening process with sChoK as a target enzyme to discover novel and effective inhibitors against sChoK. The discrepancy between the strength of sChoK inhibitory activity of quercetin and its antimicrobial activity suggests that quercetin may not easily cross the cell membrane of *S. pneumoniae* cells. However, with a model of the binding site between quercetin and sChoK, quercetin could be modified to both bind the enzyme more effectively as well as add moieties to the compound to membrane crossover.

### 2.3. Quercetin Is a Competitive Inhibitor of sChoK

In order to gain some insight into the binding model of quercetin to sChoK, kinetic assays performed on the purified recombinant sChoK using the LDH/PK method to determine the mechanism of action of quercetin. The two natural substrates of sChoK are choline and ATP. The concentration of quercetin was fixed at its IC_50_, and increasing concentrations of choline (Figure 2A) or ATP (Figure 2B) were added to determine their effects on the %inhibition of quercetin. Increasing the concentration of choline led to a decrease in %inhibition, indicating a competitive mode of inhibition with respect to choline (Figure 2A). On the other hand, increasing the concentration of ATP had no effect on %inhibition, which is consistent with a non-competitive mode of inhibition. Several kinetic constants were also measured (Figure 2C): K_I_, α, K_m_, and V_max_. Interestingly, a K_I_ (strength of binding) of 3.86 +/− 1.25 corresponded well with the measured IC_50_ of quercetin. The α-value is an index competitivity: the higher α, the stronger the competitivity of an inhibitor with the substrate being assayed, with an α reaching infinity being indicative of a purely competitive mode of inhibition; an α = 1 reveals a non-competitive mode of action, and an α-value between 1 and infinity being indicative of a somewhat competitive (mixed) mode of inhibition. With respect to choline, an α-value > 1 was measured, which was indicative of a mixed mode of inhibition, which was consistent with a Km_inh_/Km_free_ of >1 and a Vmax_inh_/V_max_ free of <1. Meanwhile, an α-value of 1 was measured in the case of ATP, which was indicative of a non-competitive mode of action, a conclusion supported by a Km_inh_/Km_free_ of ~1 and a Vmax_inh_/V_max_ free of <1.

Taken together, these results suggest that quercetin physically enters the choline binding site and displaces the choline substrate, while leaving the ATP binding site in the catalytic site untouched. Using this information, we defined the binding site of querecetin as those residues which interact with choline in a docking study of quercetin into the sChoK binding site. As shown in the most energetically favorable structure that emerged from this study, quercetin is modelled to have a binding mode that overlaps in part with the choline binding site. Therefore, quercetin can be considered a classic competititve inhibitor. Moreover, the binding mode of quercetin could be used as a scaffold to rationally design inhibitors that more effectively interact with the choline binding site.

We report here an initial screening of 8 tea extracts. This screening revealed that all 8 teas contained inibitors of sChoK, and that each had some antimicrobial activity that correlated well with their inhibitory activity. The flavonoid quercetin was found to be a competitive inhibitor of sChoK. Importantly, while querecetin has been identified as a promising natural inhibitor, the full library of potential sChoK inhibitors has not yet been determined. Therefore, further studies that make use of alternative extraction and chromatagraphic methods are warranted to fully mine the inhibitory potential of these teas. In addition, this research has implications for many diseases linked to the activity of eukaryotic ChoKs including human isoforms. These diseases include cancer, rheumatoid arthritis, and malaria. Indeed, querecetin could function as a potential natural therapeutic against these diseases via inhibition of their respective ChoKs. This study represents an important initial step towards establishing a discovery pipeline for natural ChoK inhibitors with a wide range of therapeutic applications.

## 3. Methods and Materials

All reagents were purchased from Sigma-Aldrich except for brain–heart infusion (BHI) broth, which was purchased from BD-Biosciences.

### 3.1. Production of Tea Extracts

Three grams of each type of tea leaf were steeped individually in 150 mL of boiling deionized water and was infused overnight for 16 h. The extracts were then filtrated with a 0.22 µM MCE syringe filters (Millipore, Bedford, MA, USA)*,* concentrated in an Eppendorf Vacufuge Concentrator at room temperature for 24 h and freeze-dried for 2 days (Labconco). Samples were stored at 4 degrees and then dissolved in deionized water and filtered with a 0.22 µM before use.

### 3.2. Determining IC_50_ Colorimetrically

The ability of the tea extracts to inhibit sChoK enzymatic activity was assayed in mixtures containing 8 mg/mL tea extracts and BL21 (DE3) cell extracts containing sChoK. A colorimetric method was used to quantify Cho consumption and PCho production as described previously [20].

### 3.3. Determining Minimum Inhibitory Concentration (MIC) by Broth Culture

Inhibitory concentrations of both tea extracts and quercetin were determined as previously described [21].

### 3.4. Measuring Inhibitory Activity of Tea Extracts by Plating

One hundred µL volume holes were then made into the Mueller–Hinton Plates agar plates inoculated with 5 mg of each tea extract was dissolved in 100 µL of deionized water, these mixtures were used to fill the holes made in the plates. Plates were then incubated overnight at 37 °C. A qualitative assessment was made the next day of the strength of inhibition, according to the relative diameters of the inhibition zones.

### 3.5. Recombinant sChoK Expression and Purification

sChoK was expressed and purified as described previously [21].

### 3.6. Colorimetric Method of Detection for Measuring IC_50_ of Plant Extracts 

The ability of the tea extracts to inhibit choline kinase enzymatic activity was assayed in the mixture’s tea extracts and BL21 (DE3) cell extracts containing either human or bacterial choline kinase. A colorimetric was used to quantify choline consumption and phosphocholine production as described previously [22].

### 3.7. Determining Querecetin Competitivity by LDH/pK

The mechanism of action of the inhibitors with respect to each substrate was determined as previously described using the PK/LDH method as the method of detection [23]. Briefly, using the concentration of one substrate was kept constant at the K_m_ and IC_50_ and initial velocities were measured using serial dilutions of the second substrate. The (%I) was calculated by dividing each initial velocity in the absence of compound with its corresponding velocity in the absence of compound. %I versus concentration and fitted into the following equation %I = 100 × (([I]/Ki + [S] [I]/α KsKi/(1 + [S]/Ks + [I]/Ki + [S] [I]/α KsKi))
where [S] is substrate concentration [I] is inhibitor concentration, Ks and Ki are the substrate and inhibitor dissociation constants, respectively, and α is a constant that reflects the relationship of the substrate and inhibitor.

### 3.8. Docking of Querecetin onto the Crystal Structure of sChoK

Using the information from the kinetic data indicating the competitive nature of quercetin with respect to choline, quercetin was docked into choline binding site of apo-sChoK structure as a model (RCSB accession #4R77) using the Patchdock server. The quercetin .SDF structure file was downloaded from the PubChem database [24] and converted to a .pdb using the online SMILES translator for the docking [25].

## Figures and Tables

**Figure 1 antibiotics-11-01272-f001:**
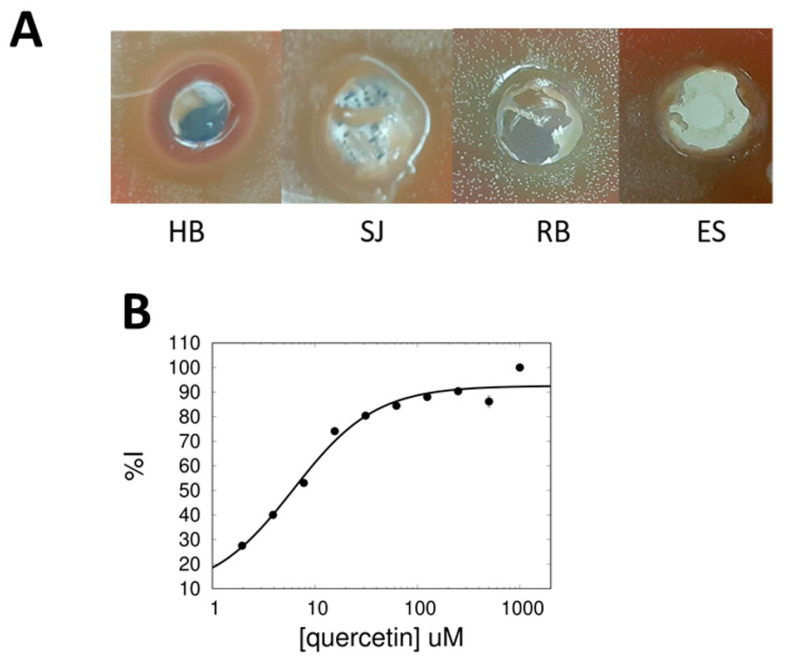
(**A**) Growth inhibition profiles from plate assays with hibiscus (HB), Saint John’s wort (SJ), Rooibis (RB) and Espinheira Santa (ES). (**B**) Dose–response assay of sChoK inhibition with quercetin.

**Table 1 antibiotics-11-01272-t001:** Tea extracts were tested for inhibitory activity against the sChoK enzyme and against *S. pneumoniae* growth. The tea extracts are ranked in order of increasing inhibitory strength against the sChoK enzyme, the strongest being Rooibos (RB), Hibiscus (HB) and Saint John’s Wort (SJ). Inhibitory strength was tested at a tea concentration of 7 mg/mL. All teas were also tested by serial dilution in broth cultures and by single dilution on Mueller–Hinton Plates (See Figure 2B).

Tea Extract	Known Antimicrobial Activity	%Inhibition sChoK	MIC *S. pneumonia*
Espinheira Santa (ES)	*S. mutans* [11] *S. aureus* [11]	17.5	40 mg/mL
Peppermint (PM)	*S. aureus* [12] *S. pyogenes* [12] *K. pneumonia* [12]	23.2	42 mg/mL
Lemon Broth (LB)	*Shigella sonei* [13]	27.6	25 mg/mL
Wormwood (W)	*S. aureus* [13]	34	17.5 mg/mL
Rose Hips (RH)	*S. aureus* [14]	35.5	60 mg/mL
Rooibos (RB)	*S. aureus* [12] *S. epidermis* [15]	44.2	5 mg/mL
Hibiscus (HB)	*S. aureus* [16]	54	4.5 mg/mL
Saint John’s Wort (SJ)	*B. cereus* [17] *B. subtilis* [18]	100	3.5 mg/mL

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
