# Peer review of "Quercetin Is a Novel Inhibitor of the Choline Kinase of Streptococcus pneumoniae"

_antibiotics, 2022, doi:10.3390/antibiotics11091272_

Round 1
Reviewer 1 Report
Comments to the authors:
1. In the title “kinase.” Needs to correct properly
2. In line 7: needs to provide correspondence details
3. In line 16: “S. pneumonia” needs to expands properly in italic, and in the remaining part of the manuscript
4. References needs to be in the journal format "Antibiotics"
5. In line 35: “int” needs to correct properly
6. In line 66: “strongest sChoK inhibition also had the highest measured MICs”, highest measured MICs or lowest measured MICs? Needs to correct properly
7. In Table: authors need to mention the tested concentration of each extract against S. pneumoniae growth.
8. In line 85: The authors need to present scientific relevant information for the presence of quercetin in their tested extracts, for example HPLC of each extract etc.
9. In line 190: “IC50”, needs to correct properly
10. In line 191: “weed extracts?”, or tea extracts
Author Response
We thank the reviewer for their careful reading of the manuscript. We have made most of the corrections as suggested by the reviewer.
Nevertheless in the case of point 8, we chose to test quercetin based on the literature suggesting that it was an abundant flavanoid found in extracts of these teas "One abundant flavonoid in the HB, SJ, and RB extracts is quercetin. Indeed, quercetin is the most abundant flavonoid in HB18 ". We proceeded with our study based on this information.
Reviewer 2 Report
The manuscript described tea extracts as natural sources of bacterial ChoK inhibitors.
The topic is of high practical interest, since commercial antibiotics are an increasing problem, intensified by the lack of new therapeutic agents.
The work presented here includes two parts.
In the first one, the authors tested the inhibitory activity of tea extract against the sChoK enzyme and S. pneumoniae growth. The results indicated that all teas contained compounds with inhibitory potential against sChoK with the most promising tea extract were SJ>HB>RB. It was demonstrated an interesting result about the correlation of MICs and inhibition percentage of the tea extracts. This inhibition was also confirmed in the plating assay.
In the second part, the authors tested quercetin, an abundant flavonoid present in HB, SJ and RB extracts. The flavonoid quercetin was demonstrated as a competitive inhibitor of sChoK.
Despite the full library of potential sChoK inhibitor has no yet been determined, this work as pioneering study provides the basis for the next line of research. So, I think the authors will have the opportunity to make these tests in their future work.
I have no comments to their experimental design or logic of the work, manuscript is clearly written and the results are well ilustrated, analyzed and discussed. To sum up, in my opinion, this is an excellent work.
I can recommend the manuscript for publication.
*MINOR POINT
The authors report (line 114 the kinetic study of quercetin versus choline and ATP substrates as Figure 2A and 2B while in Legend (line 153) was Figure 4C1 and 4C2.
The same for kinetic constants described (line 119) as figure 2C while in Legend (line 158) as Figure 4C3. Having said that, kindly adjust the text and legend.
Author Response
We thank the reviewer for their positive comments and for their careful reading of the manuscript. We have corrected the legends, as suggested.